# Sex- and nationality-based participation and performance trends in the Swissman Xtreme Triathlon (2019–2025)

Luciano Bernardes Leite[1], Pedro Forte [2,3,4], Marilia Santos Andrade[5], Sasa Duric[6], Pantelis T. Nikolaidis[7], Mabliny Thuany[8], Katja Weiss[9], Thomas Rosemann[9], Beat Knechtle[9,10]*

1 Department of Physical Education, Federal University of Viçosa, Viçosa, Brazil, 2 Department of Sports, Higher Institute of Educational Sciences of the Douro, Penafiel, Portugal, 3 Department of Sports Sciences, Instituto Politécnico de Bragança, Bragança, Portugal, 4 Research Center for Active Living and Wellbeing, Instituto Politécnico de Bragança, Bragança, Portugal, 5 Department of Physiology, Federal University of São Paulo, São Paulo, Brazil, 6 Liberal Arts Department, American University of the Middle East, Egaila, Kuwait, 7 School of Health and Caring Sciences, University of West Attica, Athens, Greece, 8 Sports Department, Physical Education, State University of Pará, Belém, Brazil, 9 Institute of Primary Care, University of Zurich, Zurich, Switzerland, 10 Medbase St. Gallen Am Vadianplatz, St. Gallen, Switzerland

* beat.knechtle@hispeed.ch

## Abstract

### Background

Participation and performance trends are well investigated for the IRONMAN® triathlon. For Xtreme Triathlons (XTri World Tour) races, only one study has examined participation and performance trends for the 'Norseman Xtreme Triathlon' in Norway, but not for other XTri World Tour events. Therefore, the aim of the present study was to investigate participation and performance trends in the 'Swissman Xtreme Triathlon' as part of the XTri World Tour.

### Methods

Finisher data from all 'Swissman Xtreme Triathlon' editions (2019–2025) were analyzed. DNS, DNF, missing information, and implausible finishing times were excluded. Participation patterns were described by sex and nationality. Sex differences in race time were evaluated using Mann–Whitney U tests. Differences among the ten most represented nationalities were tested using Welch's ANOVA with Dunnett's T3 post-hoc comparisons. Temporal changes in performance were assessed with quantile regression at the 0.25, 0.50, and 0.75 quantiles ($p < 0.05$).

### Results

A total of 1,032 finishers were included, of whom 13.5% were women. Switzerland had the highest participation (n = 431). Performance was similar across most

**Data availability statement:** All relevant data are within the paper and its Supporting information files.

**Funding:** The author(s) received no specific funding for this work.

**Competing interests:** The authors have declared that no competing interests exist.

nationalities, with slower times observed only among athletes from the United States compared with Switzerland (p = 0.01), Germany (p = 0.02), and Norway (p = 0.03). No sex-based differences were found in any edition (overall p = 0.4922; r = −0.02). Quantile regression revealed clear temporal changes in performance. At the median (0.50), race time increased by 715 s·year$^{-1}$ (95% CI: 434–997; p < 0.0001), and a similar rise occurred at the 0.75 quantile (β = 727 s·year$^{-1}$; 95% CI: 498–955; p < 0.0001). In contrast, the 0.25 quantile showed a smaller and non-significant increase (β = 345 s·year$^{-1}$; p = 0.0626), indicating that intermediate and slower athletes were primarily responsible for the overall temporal decline. Sex-specific analyses confirmed this pattern: significant increases at the median and 0.75 quantiles for men, and a significant increase only at the median quantile for women.

## Conclusions

Swiss athletes formed the largest portion of competitors in 'Swissman Xtreme Triathlon', while performance was comparable across most nationalities. Women and men performed similarly throughout all editions. Race times increased across years, particularly among intermediate and slower finishers. These findings provide an updated overview of participation and performance trends in this major XTri World Tour event. Future studies need to investigate more races of the XTri World Tour.

## Introduction

The IRONMAN® triathlon is highly popular [1,2] with both age group [3,4] and professional triathletes [5,6] competing in numerous races worldwide to qualify for the IRONMAN® World Championship in IRONMAN® Hawaii [7,8], which is considered one of the ten sport monuments [9].

Among IRONMAN® triathletes, the majority of participants are recreational age group triathletes [3], who spent a higher quantity of time and resources to improve performance outcomes. Recently, the origins of IRONMAN® age group athletes have been thoroughly investigated. Although most successful IRONMAN® age group triathletes originated from the USA, the United Kingdom, Australia and Canada [10,11], European IRONMAN® age group triathletes have been the fastest [3,10,11]. In this context, European countries such as Germany, Austria, Denmark, Belgium, Switzerland, Norway, Finland, Czechia, Estonia, and Slovenia have the fastest competitors [3,11]. A limitation of these analyses is that they have considered all IRONMAN® age group athletes from a country, but not specific age groups.

In addition to the traditional IRONMAN® races, there are also IRONMAN®-distance triathlons such as the 'Norseman Xtreme Triathlon' [12] where athletes have to face challenging natural environments (swimming in cold water), featuring long distances, steep elevation (cycling and running on hilly courses in the cold), and technical terrain, designed to test both endurance and resilience beyond traditional triathlons [13]. Regarding the origin of athletes competing in the 'Norseman Xtreme Triathlon', most

finishers have been from Norway, and the fastest race times were achieved by Norwegian women and men [14]. Studies assessing performance trends showed that from 2003 to 2015, the number of successful women increased, and women improved their performance and achieved similar performances to men in swimming, cycling and overall race time [12]. It has been noted a sex bias in human performance research of limited information on women indicating the existence of an important research gap [15].

The 'Norseman Xtreme Triathlon' is part of Xtreme Triathlons series of the XTri World Tour with an increasing high number of different extreme triathlon races held over the world [16]. While we now have some limited knowledge with the 'Norseman Xtreme Triathlon' about one race of the XTri World Tour, we lack information about participation and performance trends in other races of this XTri World Tour, such as the 'Swissman Xtreme Triathlon' held in Switzerland and the 'CELTMAN! Extreme Scottish Triathlon' held in Scotland. These races are qualifiers for 'Norseman Xtreme Triathlon' which is the World Championship of the XTri World Tour.

The aim of the present study was to investigate participation patterns and performance trends in the 'Swissman Xtreme Triathlon', one of the major events of the XTri World Tour. Specifically, we sought to examine (i) nationality-based participation profiles, (ii) sex differences in overall and split-discipline performance, and (iii) how race performance has evolved over the years, including whether temporal changes differ across the distribution of finishing times. Based on previous findings from similar extreme triathlon events such as 'Norseman Xtreme Triathlon', we hypothesized that Swiss athletes would be the most numerous competitors and that women would achieve performances comparable to those of men.

## Method

### Ethical approval

This study was approved by the Institutional Review Board of Kanton St. Gallen, Switzerland, with a waiver of the requirement for informed consent of the participants, as the study involved analysis of publicly available data (EKSG 01/06/2010). The study was conducted in accordance with recognized ethical standards according to the Declaration of Helsinki adopted in 1964 and revised in 2013.

### The race

The 'Swissman Xtreme Triathlon' is a triathlon event covering the IRONMAN® distance (3.86 km swimming, 180.25 km cycling, and 42.195 km running), held across the Swiss cantons of Ticino, Uri, Valais, and Bern. The race has been organized annually in June since 2013. The swim starts in Lake Maggiore (Ticino) in front of the Brissago Islands and covers 3.86 km to Ascona. The 180.25 km cycling segment, with 3,770 meters of elevation gain, goes from Ascona through Bellinzona over the three Alpine passes of Gotthard (2,107 m above sea level), Furka (2,429 m), and Grimsel (2,164 m) to Brienz (Bern). The final marathon run, with 1,990 meters of elevation gain, starts in Brienz and ends at Kleine Scheidegg (2,061 m). The 'Swissman Xtreme Triathlon', together with the 'Norseman Xtreme Triathlon' (Norway) and 'CELTMAN! Extreme Scottish Triathlon' (Scotland), forms the Xtreme Triathlons series (XTri World Tour) [16]. The first and second place finishers qualify for the XTri World Championships at the 'Norseman Xtreme Triathlon' in Norway. The course records for the 'Swissman Xtreme Triathlon' are 11:23 h:min, set in 2018 by Michał Rajca for men, and 12:39 h:min, set in 2013 by Emma Pooley for women.

### Data set and data preparation

Race data from all official 'Swissman Xtreme Triathlon' events were obtained from the official website [17] and included information on sex, age groups in 5-year intervals, nationality, year of participation, and both split times and overall finishing times (swimming, cycling, running, and transition times). Race data were available for the 2019, 2022, 2023, 2024, and 2025 editions. The 2013–2018 editions and years without available or complete records were excluded due

to incomplete or inconsistently reported information regarding key variables such as sex and nationality. In the earlier editions, the official results were presented in non-standardized formats, often as formatted reports rather than structured datasets, which limited reliable extraction of demographic variables. To ensure data consistency and comparability across participants, only editions with standardized and complete records were included in the analysis. Only athletes who officially completed the 'Swissman Xtreme Triathlon' were included. Records corresponding to DNS (did not start), DNF (did not finish), or missing essential information (*e.g.,* missing overall time) were removed. Relay teams were not part of the 'Swissman Xtreme Triathlon' race format and therefore did not appear in the dataset. Data cleaning procedures included the removal of implausible finishing times. As the 'Swissman Xtreme Triathlon' typically takes between 12 and 20 hours to complete, records far outside this range were considered errors and excluded from the dataset. Only a few such cases were identified and removed.

## Statistical analysis

Descriptive statistics were calculated for all study variables. Finishing times are presented primarily as medians due to their skewed distribution, while categorical variables (sex, nationality, and year of participation) are summarized as counts and percentages. Differences in performance between women and men were evaluated using the non-parametric Mann–Whitney U test, accompanied by effect size measures to assess the magnitude of observed differences. Comparisons across nationalities were conducted only for countries with at least 10 race participations. Because group sizes were unbalanced and variances differed, Welch's ANOVA was applied, followed by Dunnett's T3 for post-hoc comparisons. Temporal changes in performance were examined using quantile regression, which allows the investigation of how different segments of the performance distribution (faster, median, and slower athletes) change over time. This method was chosen because it provides insights beyond the average trend and is particularly appropriate for endurance sports data, where performance can vary substantially between athletes and across years. The significance level was set at $p < 0.05$. All statistical analyses were performed using the Statistical Package for the Social Sciences (SPSS Statistics 30; IBM, Armonk, NY, USA).

## Results

### Sample characterization

Between 2019 and 2025, a total of 1,032 race participations were recorded in the 'Swissman Xtreme Triathlon', of which 139 (13.5%) were female and 893 (86.5%) were male. Annual entries ranged from 177 in 2024–247 in 2025. Male participation consistently exceeded female participation, with women representing between 12% and 16% of each cohort. Despite variations in total participations across years, the relative proportion of female entries remained stable, highlighting a persistent underrepresentation of women in the event. Fig 1 illustrates the distribution of race participations by sex from 2019 to 2025.

Switzerland showed the highest number of race participations (n = 431), followed by France (n = 94), Germany (n = 69), and Italy (n = 42). Other European countries such as the United Kingdom (n = 36), Norway (n = 33), Belgium (n = 30), and the Netherlands (n = 29) also had notable representation. Outside Europe, Poland (n = 21) and the United States (n = 21) completed the group of the ten most represented nationalities. Importantly, these values refer to race participations, not unique athletes, as some competitors may have appeared in multiple editions. Fig 2 summarizes the distribution of participations for the top 10 nationalities.

Fig 3 shows the performance values for the ten countries with the largest sample sizes. Welch ANOVA indicated significant differences between groups (Welch's $F (9, 110.53) = 4.00$, $p = 0.00019$). Post-hoc comparisons using Dunnett's T3 revealed that only the United States differed significantly from other countries, presenting higher mean times and therefore worse performance when compared with Switzerland (p = 0.01), Germany (p = 0.02), and Norway (p = 0.03). No other pairwise differences were statistically significant.

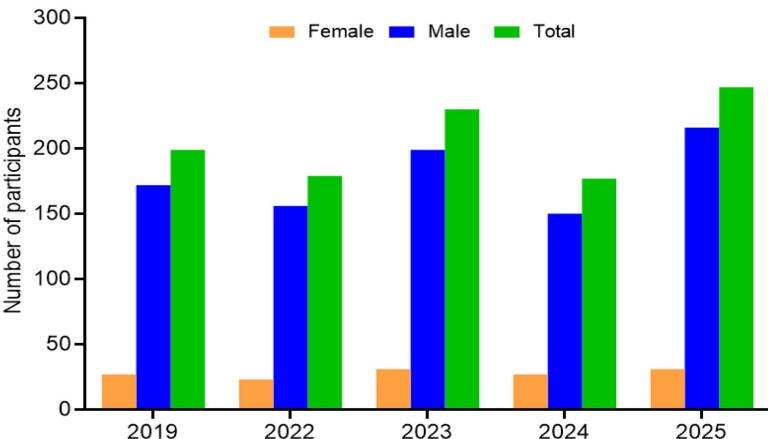

**Fig 1. Number of participants per year in 'Swissman Xtreme Triathlon'.**

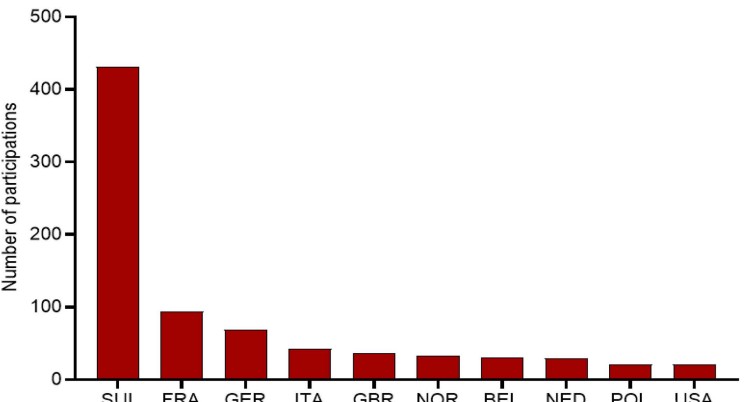

**Fig 2. Number of race participations per nationality for the ten most represented countries in the 'Swissman Xtreme Triathlon'.** Values represent total race participations across all analyzed editions.

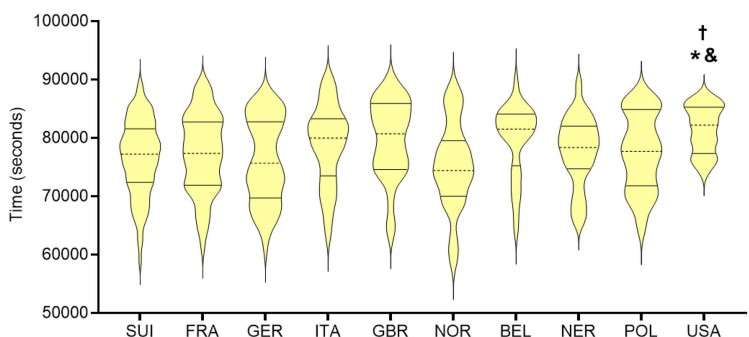

**Fig 3. Distribution of finishing times (seconds) for athletes from the ten countries with the largest number of participants in the 'Swissman Xtreme Triathlon'.** Violin plots represent the distribution of finishing times for each nationality. Statistical differences were assessed using Welch ANOVA followed by Dunnett's T3 post-hoc test. * $p < 0.05$ vs. SUI; # $p < 0.05$ vs. GER; † $p < 0.05$ vs. NOR.

Fig 4 presents the comparison between male and female athletes across all editions analyzed. Across all editions from 2019 to 2025, no significant differences were observed between male and female athletes in total race time. The Mann-Whitney tests were non-significant for every year analyzed, with p-values of 0.6432 (2019), 0.9659 (2022), 0.9595 (2023), 0.3535 (2024), and 0.2256 (2025). Consistently, the effect sizes were trivial in all cases (Rosenthal's r ranging from −0.00 to −0.08). When all years were combined, the overall comparison also showed no sex-based difference (p = 0.4922; r = −0.02). Median race times were similar across all analyses, and the small magnitude of differences reinforces that both men and women performed comparably throughout the entire period.

Across all athletes (Table 1) and years (2019–2025), quantile regression showed that total race time increased over time, particularly at the median and upper quantiles. At the 0.50 quantile (median athletes), each additional year was associated with an increase of 715 s in overall race time (95% CI: 433.8 to 996.5; p < 0.0001), while similar effects were

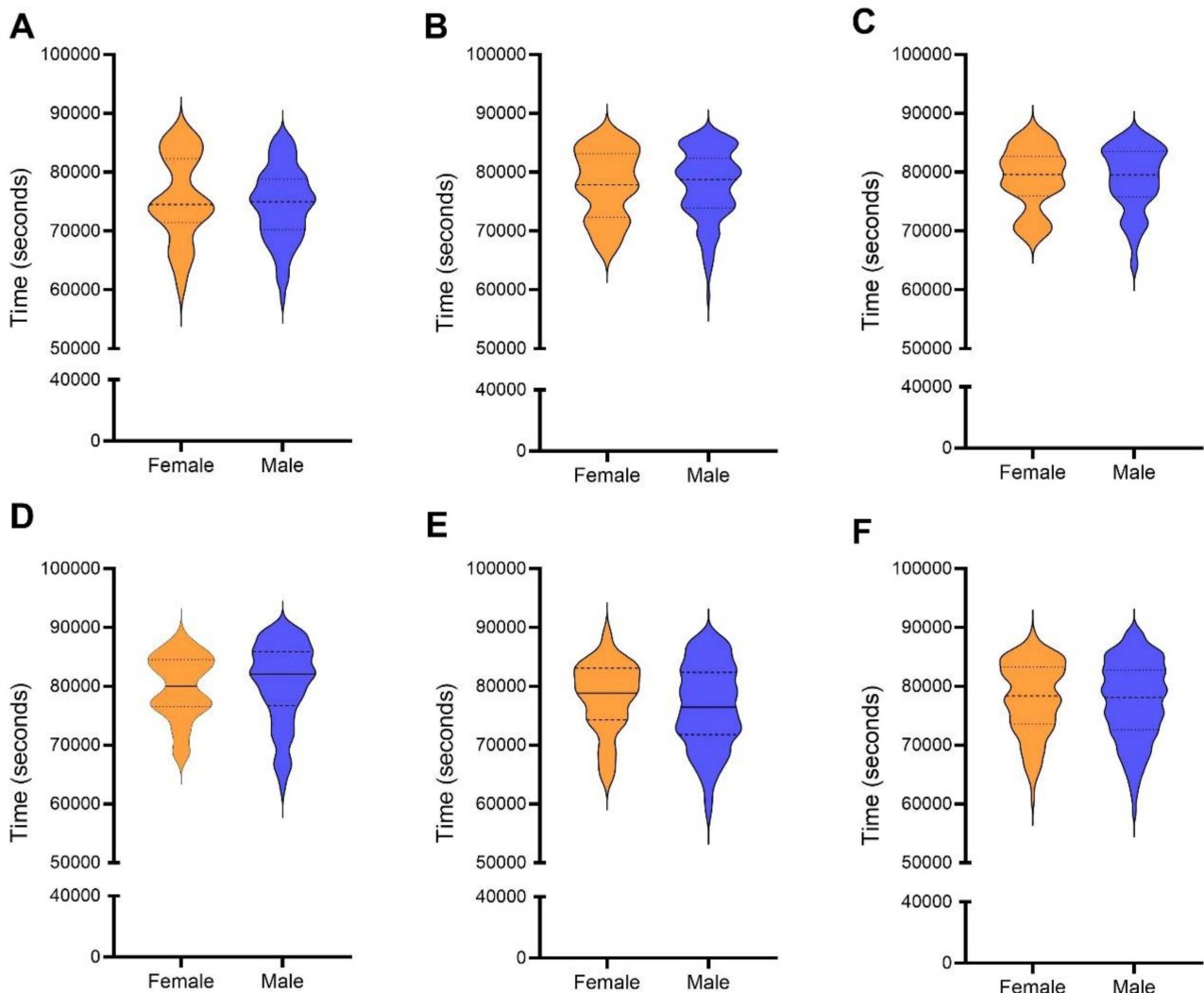

**Fig 4. Violin plots showing the distribution of total race time (in seconds) for female and male athletes in 'Swissman Xtreme Triathlon'.** (A) 2019 edition; (B) 2022 edition; (C) 2023 edition; (D) 2024 edition; (E) 2025 edition **(F)** All years combined. Mann–Whitney test.

**Table 1. Quantile regression estimates for race time across years (all athletes).**

| Quantile | β (s/year) | 95% CI β (s/year) | p | Pseudo R² |
|---|---|---|---|---|
| 0.25 | 344.67 | −18.20 to 707.53 | 0.0626 | −0.247 |
| 0.50 | 715.17 | 433.80 to 996.54 | <0.0001 | 0.025 |
| 0.75 | 726.83 | 498.30 to 955.37 | <0.0001 | −0.161 |

Note. CI = confidence interval; Q = quantile; Pseudo R² based on absolute deviation method.

observed at the 0.75 quantile (slower athletes) (β = 726.8 s·year$^{-1}$; 95% CI: 498.3 to 955.4; p < 0.0001). At the 0.25 quantile (faster athletes), the slope was smaller and did not reach statistical significance (β = 344.7 s·year$^{-1}$; 95% CI: −18.2 to 707.5; p = 0.0626), suggesting that performance changes were more evident among intermediate and slower athletes.

In male (Table 2), the median quantile showed a significant increase in total race time over the years (β = 675.2 s·year$^{-1}$; 95% CI: 366.2 to 984.2; p < 0.0001), with a similar pattern at the 0.75 quantile (β = 825.3 s·year$^{-1}$; 95% CI: 579.5 to 1071.2; p < 0.0001). No significant trend was observed at the 0.25 quantile (p = 0.1321).

In female (Table 3), a significant temporal increase in total race time was observed at the median quantile (β = 841.7 s·year$^{-1}$; 95% CI: 96.9 to 1586.5; p = 0.0271), whereas trends at the 0.25 and 0.75 quantiles did not reach statistical significance (p = 0.0854 and p = 0.5704, respectively).

## Discussion

The purpose of this study was to examine participation patterns and performance trends in the 'Swissman Xtreme Triathlon', with particular focus on nationality profiles, sex differences, and how race performance evolved across recent years. The main findings revealed that Swiss athletes were indeed the most numerous, although their performance did not differ meaningfully from that of other European nationalities, with only athletes from the United States showing consistently slower finishing times. Women and men demonstrated remarkably similar race performances across all editions,

**Table 2. Quantile regression model evaluating temporal trends in male performance.**

| Quantile | β (s/year) | 95% CI β (s/year) | p | Pseudo R² |
|---|---|---|---|---|
| 0.25 | 300.00 | −90.69 to 690.69 | 0.1321 | −0.249 |
| 0.50 | 675.17 | 366.18 to 984.15 | <0.0001 | 0.023 |
| 0.75 | 825.33 | 579.50 to 1071.16 | <0.0001 | −0.167 |

Note. CI = confidence interval; Q = quantile; Pseudo R² based on absolute deviation method.

**Table 3. Quantile regression model evaluating temporal trends in female performance.**

| Quantile | β (s/year) | 95% CI β (s/year) | p | Pseudo R² |
|---|---|---|---|---|
| 0.25 | 700.83 | −99.16 to 1500.82 | 0.0854 | −0.202 |
| 0.50 | 841.67 | 96.86 to 1586.47 | 0.0271 | 0.043 |
| 0.75 | 191.83 | −476.01 to 859.67 | 0.5704 | −0.197 |

Note. CI = confidence interval; Q = quantile; Pseudo R² based on absolute deviation method.

supporting the notion that sex differences tend to diminish under conditions of extreme endurance. Additionally, race times increased progressively over the years, particularly among intermediate and slower athletes, suggesting that temporal factors, environmental variability, and shifting participant profiles may have influenced overall performance. These findings collectively provide new insight into participation dynamics and performance development within one of the flagship events of the XTri World Tour.

### National representation and performance comparisons

The participation in the 'Swissman Xtreme Triathlon' was largely dominated by Swiss athletes, followed by competitors from France, Germany, and Italy. This distribution is likely influenced by the location of the event, which naturally favors athletes from European countries—especially Switzerland—due to proximity and easier logistical access. Moreover, the geographical characteristics and higher levels of economic development in these regions may positively influence access to training infrastructure, competition opportunities, and long-term athletic development [18]. The economic dimension is particularly important in this format of triathlon, as elite athletes must invest substantial amounts not only in scientific support, sports equipment, and nutritional aids, but also in extensive travel, often abroad [19].

When analyzing race times among the most represented nationalities, it was observed that, despite small variations in the averages, most countries showed broadly similar performances. The apparent differences were not sufficiently consistent to indicate a clear advantage for any European nation across the editions. The only exception observed was among athletes from the United States, who displayed higher times compared to some of the most present European countries, such as Switzerland, Germany, and Norway. This isolated difference may reflect factors such as lower familiarity with alpine courses, differences in specific preparation, or distinct profiles of athletes who choose to compete internationally in this type of event.

### No significant race time differences between women and men

The first and most important finding was that the global comparison of overall race times revealed no statistically significant differences between the sexes, confirming our hypothesis based on findings from the 'Norseman Xtreme Triathlon' where women achieved a similar performance to men in swimming, cycling and overall race time [12]. It is possible that female triathletes can achieve performances comparable to male triathletes under extreme conditions. For example, women competing in the 2024 Quintuple Ultra Triathlon World Championship in Colmar, France, covering 19 km swimming, 900 km cycling and 211 km running, were able to outperform men in swimming and cycling [20]. The most likely explanation is the elite nature of the World Championship, which featured a highly selected and committed female cohort with a high completion rate. Furthermore, it was the largest and fastest World Championship in Quintuple Iron ultra-triathlon history [20].

It has been suggested that there was sex difference in human performance during competitions characterized by increased demands in strength, speed, power and endurance, where men outperformed women mainly due to the direct and indirect effect of hormones [21]. The male advantage in endurance-related performance has been mostly due to their superior maximal oxygen uptake levels either in absolute or relative to body mass values [22]. On the other hand, these differences might be attenuated by certain advantages of women such as a larger percentage of Type I skeletal muscle fibers and myosin heavy chain I [23].

In addition, physiological, metabolic and strategic factors that favor women in extreme endurance events may also contribute. Women exhibit a higher proportion of type I muscle fibers, greater resistance to muscular fatigue, and more efficient utilization of fatty acids as an energy substrate, which preserves muscle glycogen during prolonged exercise. This results in a smaller relative decline in speed with increasing distance, whereas men tend to show a greater performance decrement in very long events [24]. Furthermore, women often adopt more consistent pacing strategies and experience lower post-exercise muscle damage, contributing to more stable performance in ultra-endurance competitions [25].

In events such as ultra-marathons and long-distance swimming, the sex difference in finish times can be reduced to ~1–4%, particularly when the number of male and female participants is similar [26,27]. The American College of Sports Medicine emphasizes that, despite higher absolute $VO_2$max values in men, the relative limiting mechanisms related to sustained intensity are similar between sexes in long-duration events [27].

### Temporal trends in performance

Across the period analyzed, overall race times showed a clear tendency to increase, indicating that athletes took progressively longer to complete the 'Swissman Xtreme Triathlon'. Although year-to-year variation is expected in events of this magnitude, the pattern observed suggests a gradual decline in performance rather than random fluctuations. This trend was most evident among athletes situated in the middle and upper portions of the performance distribution, while those at the fastest end remained relatively stable across editions. Such divergence implies that intermediate and slower participants may have been disproportionately affected by the conditions or demands of the race over time.

### Limitations, implications for future research, and practical applications

Due to incomplete race results in earlier years, we could not include all race results since the race began. Several early editions (2013–2018) reported results in non-standardized formats and lacked consistently structured information for key variables such as sex and nationality, which prevented reliable data extraction and comparison across participants. Including more years might have led to different results, with men possibly being faster than women overall. In addition, the analysis relied on publicly available race data that lacked key training-related variables (*e.g.,* altitude gain, endurance background, prior performance) [4,28] as well as environmental characteristics such as competition location, average temperature, humidity, wind, and elevation gain [29,30]. These factors could have provided important context regarding race conditions and performance constraints. Finally, we were unable to control for repeated performances by the same athlete, which may have introduced a small degree of bias. Future studies could analyze race results from the 'CELTMAN! Extreme Scottish Triathlon' held in Scotland, where women might also outperform men. For athletes and coaches, our findings suggest that women may have the opportunity to achieve performances comparable to, or in some cases surpassing, those of men in XTri World Tour events. Since women are able to achieve similar or even better performances in triathlons under extreme conditions, some women may wish to consider extreme triathlon events as an alternative to conventional IRONMAN® races, particularly if they are attracted to longer and more demanding courses.

## Conclusion

Participation in the 'Swissman Xtreme Triathlon' increased over the years, with Swiss athletes forming the largest group of competitors. Performance was similar across most nationalities, except for slower times observed among athletes from the United States. Women and men showed comparable race performances, and overall finishing times gradually increased across editions, especially among intermediate and slower athletes. These findings clarify recent participation patterns and performance developments in this major XTri event. Future studies need to investigate more races of the XTri World Tour.

## Supporting information

**S1 File. PONE-D-25–65637_Swissman.**
(XLSX)

## Author contributions

**Conceptualization:** Beat Knechtle.

**Data curation:** Beat Knechtle.

**Formal analysis:** Luciano Bernardes Leite, Pedro Forte.

**Writing – original draft:** Luciano Bernardes Leite, Pedro Forte.

**Writing – review & editing:** Marilia Santos Andrade, Sasa Duric, Pantelis T. Nikolaidis, Mabliny Thuany, Katja Weiss, Thomas Rosemann, Beat Knechtle.

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
