## [Decision Letter · Decision Letter 0]

5 Mar 2026

Sex- and nationality-based participation and performance trends in the Swissman Xtreme Triathlon (2019–2025)

PONE-D-25-65637

Dear Dr. Knechtle,

We’re pleased to inform you that your manuscript has been judged scientifically suitable for publication and will be formally accepted for publication once it meets all outstanding technical requirements.

Kind regards,

Ratko Peric, PhD

Academic Editor

PLOS One

**Journal requirements:** 1. Thank you for uploading your study's underlying data set. Unfortunately, the repository you have noted in your Data Availability statement does not qualify as an acceptable data repository according to PLOS's standards. At this time, please upload the minimal data set necessary to replicate your study's findings to a stable, public repository (such as figshare or Dryad) and provide us with the relevant URLs, DOIs, or accession numbers that may be used to access these data. For a list of recommended repositories and additional information on PLOS standards for data deposition, please see https://journals.plos.org/plosone/s/recommended-repositories. 2. In the online submission form, you indicated that “The datasets used and/or analysed during the current study are available from the corresponding author on reasonable request.”All PLOS journals now require all data underlying the findings described in their manuscript to be freely available to other researchers, either a. In a public repository, b. Within the manuscript itself, or c. Uploaded as supplementary information.This policy applies to all data except where public deposition would breach compliance with the protocol approved by your research ethics board. If your data cannot be made publicly available for ethical or legal reasons (e.g., public availability would compromise patient privacy), please explain your reasons on resubmission and your exemption request will be escalated for approval. 3. Your ethics statement should only appear in the Methods section of your manuscript. If your ethics statement is written in any section besides the Methods, please delete it from any other section. 4. We note that there is identifying data in the Supporting Information file. Due to the inclusion of these potentially identifying data, we have removed this file from your file inventory. Prior to sharing human research participant data, authors should consult with an ethics committee to ensure data are shared in accordance with participant consent and all applicable local laws. Data sharing should never compromise participant privacy. It is therefore not appropriate to publicly share personally identifiable data on human research participants. The following are examples of data that should not be shared: -Name, initials, physical address-Ages more specific than whole numbers-Internet protocol (IP) address-Specific dates (birth dates, death dates, examination dates, etc.)-Contact information such as phone number or email address-Location data-ID numbers that seem specific (long numbers, include initials, titled “Hospital ID”) rather than random (small numbers in numerical order) Data that are not directly identifying may also be inappropriate to share, as in combination they can become identifying. For example, data collected from a small group of participants, vulnerable populations, or private groups should not be shared if they involve indirect identifiers (such as sex, ethnicity, location, etc.) that may risk the identification of study participants. Additional guidance on preparing raw data for publication can be found in our Data Policy (https://journals.plos.org/plosone/s/data-availability#loc-human-research-participant-data-and-other-sensitive-data) and in the following article: http://www.bmj.com/content/340/bmj.c181.long. Please remove or anonymize all personal information (<specific identifying information in file to be removed>), ensure that the data shared are in accordance with participant consent, and re-upload a fully anonymized data set. Please note that spreadsheet columns with personal information must be removed and not hidden as all hidden columns will appear in the published file.

Reviewers' comments:

Reviewer's Responses to Questions

**Comments to the Author**

1. Is the manuscript technically sound, and do the data support the conclusions?

Reviewer #1: Yes

Reviewer #2: Yes

Reviewer #3: Yes

Reviewer #4: Yes

2. Has the statistical analysis been performed appropriately and rigorously?

Reviewer #1: Yes

Reviewer #2: Yes

Reviewer #3: Yes

Reviewer #4: Yes

3. Have the authors made all data underlying the findings in their manuscript fully available?

Reviewer #1: Yes

Reviewer #2: Yes

Reviewer #3: Yes

Reviewer #4: Yes

4. Is the manuscript presented in an intelligible fashion and written in standard English?

Reviewer #1: Yes

Reviewer #2: Yes

Reviewer #3: Yes

Reviewer #4: Yes

**Reviewer #1:** This manuscript presents a clear, methodologically sound, and timely analysis of participation and performance trends in the Swissman Xtreme Triathlon as part of the XTri World Tour. The study addresses a relevant gap in the literature, as evidence on Xtreme Triathlons beyond the Norseman event remains scarce.This manuscript presents a clear, methodologically sound, and timely analysis of participation and performance trends in the Swissman Xtreme Triathlon as part of the XTri World Tour. The study addresses a relevant gap in the literature, as evidence on Xtreme Triathlons beyond the Norseman event remains scarce.This manuscript presents a clear, methodologically sound, and timely analysis of participation and performance trends in the Swissman Xtreme Triathlon as part of the XTri World Tour. The study addresses a relevant gap in the literature, as evidence on Xtreme Triathlons beyond the Norseman event remains scarce.This manuscript presents a clear, methodologically sound, and timely analysis of participation and performance trends in the Swissman Xtreme Triathlon as part of the XTri World Tour. The study addresses a relevant gap in the literature, as evidence on Xtreme Triathlons beyond the Norseman event remains scarce.

The dataset is comprehensive, covering all editions of the event from 2019 to 2025, and the inclusion and exclusion criteria are appropriate and transparently described. The statistical approach is rigorous and well suited to the research questions. In particular, the use of quantile regression provides valuable insight into temporal performance changes across different performance levels, going beyond mean-based analyses and strengthening the interpretation of trends over time.

The results are clearly presented and logically interpreted. The conclusions are fully supported by the data and are appropriately framed without overgeneralization. The finding that performance declines are driven mainly by intermediate and slower finishers, while elite performance remains relatively stable, is especially informative and relevant for understanding participation dynamics in extreme endurance events.

The manuscript is well written, intelligible, and structured in accordance with journal standards. Data availability is adequate and consistent with PLOS ONE policies.

Overall, this is a solid and well-executed contribution to the literature on endurance sports and Xtreme Triathlon events. I have no major or minor concerns and recommend the manuscript for acceptance as submitted.

**Reviewer #2:** I would like to congratulate the authors on a well-prepared and clearly written manuscript. The study is methodologically sound, clearly presented, and addresses an interesting and relevant research question. I also appreciate the transparent use of publicly available data and the appropriate application of statistical methods. Overall, this manuscript represents a valuable contribution to the field, and I would like to thank the authors for their thorough and high-quality work.I would like to congratulate the authors on a well-prepared and clearly written manuscript. The study is methodologically sound, clearly presented, and addresses an interesting and relevant research question. I also appreciate the transparent use of publicly available data and the appropriate application of statistical methods. Overall, this manuscript represents a valuable contribution to the field, and I would like to thank the authors for their thorough and high-quality work.I would like to congratulate the authors on a well-prepared and clearly written manuscript. The study is methodologically sound, clearly presented, and addresses an interesting and relevant research question. I also appreciate the transparent use of publicly available data and the appropriate application of statistical methods. Overall, this manuscript represents a valuable contribution to the field, and I would like to thank the authors for their thorough and high-quality work.I would like to congratulate the authors on a well-prepared and clearly written manuscript. The study is methodologically sound, clearly presented, and addresses an interesting and relevant research question. I also appreciate the transparent use of publicly available data and the appropriate application of statistical methods. Overall, this manuscript represents a valuable contribution to the field, and I would like to thank the authors for their thorough and high-quality work.

Overall, I believe that the manuscript is well prepared, and I have no comments or suggestions for revision.

**Reviewer #3:** Manuscript Number: PONE-D-25-65637 Manuscript Number: PONE-D-25-65637 Manuscript Number: PONE-D-25-65637 Manuscript Number: PONE-D-25-65637

Manuscript Title: Sex- and nationality-based participation and performance trends in the Swissman Xtreme Triathlon (2019–2025)

This manuscript examines participation and performance trends in the Swissman Xtreme Triathlon between 2019 and 2025. It focuses particularly on differences based on sex and nationality, as well as temporal changes in race performance. This topic is highly relevant given the scarcity of research on participation and performance trends in XTri World Tour events, with prior studies being limited to the Norseman Xtreme Triathlon in Norway. The present study therefore fills an important gap by providing updated insights from one of the XTri World Tour's major races.

The manuscript is well written and logically structured, and the statistical methods used are appropriate. The results are clearly presented and largely support the authors’ conclusions. Overall, this study is a valuable addition to the literature on extreme endurance sports.

I just have a few minor comments to make.

- Overall, the different sections of the manuscript are well written.

- Some of the sentences in the introduction and discussion sections are lengthy and could be simplified for clarity.

- Consider standardising the terminology relating to 'Xtreme Triathlon', 'XTri World Tour' and 'Xtreme Triathlon Series'.

- In the Methods section, it would be helpful to briefly explain why the 2013–2018 editions were excluded, as well as mentioning any potential biases that this exclusion may have caused.

- You could add a few recent studies on ultra-endurance or extreme triathlons to better place these findings in context, which would provide a more comprehensive overview of the subject.

- The figures and tables are generally clear. However, adding short explanatory captions to some of the figures (e.g. Figures 2 and 3) would help readers to interpret the data more quickly.

**Reviewer #4:** Nice pieece of information and good selection of the topic. Please continue publishing these kind of information which adds to scientific community extra value. Probably with additional competitions and from different countries participants information will be more robust Nice pieece of information and good selection of the topic. Please continue publishing these kind of information which adds to scientific community extra value. Probably with additional competitions and from different countries participants information will be more robust Nice pieece of information and good selection of the topic. Please continue publishing these kind of information which adds to scientific community extra value. Probably with additional competitions and from different countries participants information will be more robust Nice pieece of information and good selection of the topic. Please continue publishing these kind of information which adds to scientific community extra value. Probably with additional competitions and from different countries participants information will be more robust

.

Reviewer #1: **Yes:** Prof. Robert Gajda, MD, PhD, DScProf. Robert Gajda, MD, PhD, DScProf. Robert Gajda, MD, PhD, DScProf. Robert Gajda, MD, PhD, DSc

Reviewer #2: No

Reviewer #3: **Yes:** Nejmeddine OuerghiNejmeddine OuerghiNejmeddine OuerghiNejmeddine Ouerghi

Reviewer #4: No

---

## [Editor Report · Acceptance letter]

PONE-D-25-65637

PLOS One

Dear Dr. Knechtle,

I'm pleased to inform you that your manuscript has been deemed suitable for publication in PLOS One. Congratulations! Your manuscript is now being handed over to our production team.

Kind regards,

on behalf of

Dr. Ratko Peric

Academic Editor

PLOS One